# Proteome Dynamics Analysis Reveals the Potential Mechanisms of Salinity and Drought Response during Seed Germination and Seedling Growth in *Tamarix hispida*

**DOI:** 10.3390/genes14030656

**Published:** 2023-03-05

**Authors:** Xin’an Pang, Shuo Liu, Jiangtao Suo, Tiange Yang, Samira Hasan, Ali Hassan, Jindong Xu, Sushuangqing Lu, Sisi Mi, Hong Liu, Jialing Yao

**Affiliations:** 1College of Life Science and Technology, Huazhong Agricultural University, Wuhan 430070, China; 2Key Laboratory of Protection and Utilization of Biological Resources in Tarim Basin, Xinjiang Production and Construction Corps, College of Life Sciences, Tarim University, Alar 843300, China; 3Hubei Provincial Key Laboratory for Protection and Application of Special Plant Germplasm in Wuling Area of China, College of Life Sciences, South-Central University for Nationalities, Wuhan 430074, China

**Keywords:** *Tamarix hispida*, seed germination, seedling growth, proteomics, drought stress

## Abstract

Understanding the molecular mechanisms of seed germination and seedling growth is vital for mining functional genes for the improvement of plant drought in a desert. *Tamarix hispida* is extremely resistant to drought and soil salinity perennial shrubs or trees. This study was the first to investigate the protein abundance profile of the transition process during the processes of *T. hispida* seed germination and seedling growth using label-free proteomics approaches. Our data suggested that asynchronous regulation of transcriptomics and proteomics occurs upon short-term seed germination and seedling growth of *T. hispida*. Enrichment analysis revealed that the main differentially abundant proteins had significant enrichment in stimulus response, biosynthesis, and metabolism. Two delta-1-pyrroline-5-carboxylate synthetases (P5CS), one Ycf3-interacting protein (Y3IP), one low-temperature-induced 65 kDa protein-like molecule, and four peroxidases (PRX) were involved in both water deprivation and hyperosmotic salinity responses. Through a comparative analysis of transcriptomics and proteomics, we found that proteomics may be better at studying short-term developmental processes. Our results support the existence of several mechanisms that enhance tolerance to salinity and drought stress during seedling growth in *T. hispida*.

## 1. Introduction

The growth cycle of a plant usually consists of seed germination and vegetative and reproductive growth; among the three stages, seed germination is the initial phase in a life cycle [1,2] and is the most sensitive to abiotic stress [3]. In arid and semi-arid regions, the most common problems affecting plant growth are the environmental stressors drought and salinity [4]. The phases of seed germination and subsequent seedling growth that could be particularly affected by the drought and salinity stress in many plant species [5]. A plant’s response to drought and salinity stress involves thousands of protein-coding genes and biochemical molecular mechanisms [6,7]. Research on the mechanisms of drought and salinity resistance would—for a low price—help to improve the effective use of land, but also support sustainable agriculture. Many previous studies have reported on drought and salinity stress responses of crops in regard to seed germination and seedling growth. Such crops have included *Avena sativa* [8], *Helianthus annuus* [9], *Oryza sativa* [10], and *Spinacia oleracea* [11]. In contrast, there are few reports on the adaptability of desert plants to drought and salinity stress during seed germination and seedling growth, even though desert plants are highly tolerant of water scarcity, and saline and alkaline environments, so they are the ideal materials for studying the response mechanisms of plants to the abiotic stressors drought and salinity.

One of the trees that is extremely resistant to drought and soil salinity, *T. hispida* Willd. is a woody halophyte that can form the natural forest in 1% salt desert environments. This kind of perennial shrub has high tolerance to drought and salinity, so it is an excellent material for research on drought and salinity resistance mechanisms and cloning of candidate genes. The overexpression of *ThMYB13* and *ThSOS3* can significantly improve salt resistance in *T. hispida* plants [12,13]. *ThNAC4*, *ThNAC7*, and *ThNAC12* increase plant abiotic stress tolerance by enhancing ROS scavenging ability and increasing the osmotic potential [14,15,16]. The *ThTrx5* extracted from *T. hispida* improves the salt tolerance of transgenic Arabidopsis thaliana via regulating biological and metabolic pathways [17]. Furthermore, plants can be regulated by multiple phytohormones at each stage of growth [18], such as the antagonistic action of abscisic acid and gibberellin [19] and the interaction among ethylene, jasmonates, and other phytohormones [20].

In recent years, with the rapid development of omics technology, many omics studies of seed germination and seedling growth were published, including transcriptomics [21,22], metabolomics [23], and proteomics [24] ones. In particular, the advances in proteomics have provided a powerful tool for studying plant development, and high-resolution mass spectrometry allows for systematic analysis of protein abundance and function and reveals changes in a plant’s development process [25]. Therefore, proteome research is important, as proteins are the ultimate biomolecules that determine the structures and functions of living systems. In recent years, proteomics has been applied successfully to seed germination and seedling growth in many model plants and crops, such as wheat [26,27,28], cress [29], rice [30,31], barley [32], sugar beet [33], and maize [34].

In the present study, the integrative proteome analysis during seed germination and seedling growth of *T. hispida* was performed through a comparative proteomic and transcriptomic approach. Comparative analysis of proteomic and transcriptomic data indicated the proteome is more suitable for the analysis of short-term developmental processes, such as seed germination and early seedling growth. The dynamic changes in the synthetase proteins in flavonoid biosynthesis were observed. The clustering analysis of Gene Ontology (GO) enrichment results show that seed germination and seedling growth play important roles in the formation of drought and salinity resistance, and the abundance patterns of key drought- and salinity-resistance-related differentially abundant proteins are also presented. The results provide new valuable insights into the proteomic mechanisms of germination and seedling growth and their effects on the resistance formation of desert plants.

## 2. Materials and Methods

### 2.1. Plant Material and Treatments

The matured dried seeds of the *T. hispida*, which were obtained from Alar City, Xinjiang Uygur Autonomous Region of China, were used as the material in this study. The full seeds were placed on absorbent paper filled with ultrapure water inside a 15 cm in diameter petri dish at intelligent light incubator (25 °C, 16 h illumination/8 h darkness) to allow them to germinate naturally. For protein extraction, the samples of the slow-water-absorption period (stage 3 at 5 h), hypocotyl-extension period (stage 4 at 24 h), and cotyledon-unfolding period (stage 5 at 144 h) were collected [35]. The materials are then stored in the −80 °C refrigerator for later use. In this study, each stage of sampling included 3 biological replicates. We declare that we obtained *T. hispida* seeds with permission.

### 2.2. Protein Extraction and Trypsin Digestion

The plant samples were first ground under liquid nitrogen treatment, and then ultrasound sonification was performed three times in the lysis buffer. The product was then combined with an equal volume of trisaturated phenol (pH 8.0) and further vortexed (4 °C, 10 min, 5000 g) for 5 min. At least 4 volumes of ammonium sulfate were added to the upper phenol phase with saturated methanol and placed at −20 °C for 6 h for protein precipitation. After centrifugation (4 °C, 10 min), the supernatant was removed; and the precipitate was retained, washed with ice-cold methanol once, and then washed with ice-cold acetone for three times. The obtained protein was redissolved using 8 M urea, and the protein concentration was measured using a BCA kit.

For digestion, 5 mM dithiothreitol was added to the protein solution for reduction (30 min, 56 °C), and 11 mM iodoacetamide was added for alkylation (15 min) in the dark. Then 200 mM TEAB was added to the protein sample to dilute its urea concentration (less than 2 M). Finally, 50 times the protein mass of trypsin was added for the first digestion (overnight). Then, 100 times the protein mass of trypsin was added for a second digestion (4 h). Finally, the peptide was desalted by a C18 solid-phase extraction column.

### 2.3. TMT Labeling and HPLC Fractionation

First, the tryptic peptides were dissolved using 0.5 M TEAB. Based on our specifications, we used the TMT reagent to label each channel of the peptide, and then incubated (room temperature) them for 2 h. Each sample was analyzed by mass spectrometry to check the success rate of labeling, and 5% hydroxylamine was then added to quench the sample. The obtained samples were then desalted and subjected to vacuum centrifugal drying.

Next, the obtained samples were fractionated into fractions using pH reverse-phase high-performance liquid chromatography, and the peptides were eventually separated into 60 fractions (Appendix A). Then, the obtained peptides were reassembled into six fractions and vacuum-centrifuge dried.

### 2.4. Mass Spectrometry (MS) Analysis

In step one, the tryptic peptides were separated using a reverse-phase analytical column on an EASY-nLC 1200 ultra performance liquid chromatography (UPLC) system. In step two, the isolated peptides were analyzed in a mass spectrometer (Orbitrap Exploris™ 480, ThermoFisher Scientific, Shanghai, China, electrospray voltage: 2.3 kV, scan resolution: 60,000, scan range: 400–1200 *m*/*z*) using a nanoelectrospray ion source. In step three, the most abundant 25 precursors were selected for 30 s dynamic exclusion using MS/MS analyses. Finally, the high-energy collisional dissociation fragmentation was performed, and then, obtained fragments were detected by Orbitrap (resolution: 15,000, first mass: 100 *m*/*z*, AGC target: 100%, intensity threshold: 5 × 10^4^ ions/s, maximum injection time: 30 ms).

### 2.5. Database Search

The proteome Discoverer (v2.4.1.15) [36] was used to analyze the obtained mass spectrum (MS) data. Tandem MS were aligned to transcriptome database of *T. hispida* (FDR < 1%). Protein relative quantification was performed in MaxQuant software (v.1.6.15.0) [37] for peptides.

### 2.6. Protein Function Annotation

This was based on UniProt-GOA database, we performed the Gene Ontology (GO) annotation of proteome. First was protein ID mapping to GO IDs by homologous alignment using the UniProt database. The InterProScan software (version: 5.39–77.0) was used for supplementary GO functional annotation of proteins. Then, proteins’ GO function descriptions were classified to three categories: biological process (BPs), cellular components, (CCs) and molecular function (MFs).

Domain annotation: the InterProScan software was used to identify the functional description of each protein’s domain by protein-sequence similarity.

Based on the KEGG database (https://www.genome.jp/kegg/, accessed on 17 November 2022), we performed the KEGG-pathway annotation of the identified proteins. Firstly, the KAAS was used to annotate the KEGG descriptions of proteins. Then, the KEGG mapper was used to map the proteins’ KEGG annotations to KEGG pathways [38,39,40].

### 2.7. Differential Analysis of Proteins

Firstly, the differential abundance analysis was performed between samples, including stage 3, stage 4, and stage 5. In the comparison group, the fold change (FC) was calculated using the relative quantitative mean of each protein. Then, the coefficient of variation (CV) was used to judge the significance of the difference. Proteins that met the following conditions were considered as having differential expression: differentially upregulated: FC > 1.3 and CV < 0.1; differentially downregulated: FC < 0.77 and CV < 0.1.

### 2.8. Functional Enrichment

Based on protein’s GO annotation, proteins were classified into different GO entries. Fisher’s exact test was performed to detect the significance of enrichment. If the *p*-value < 0.05, the GO entry was considered to be significantly enriched. 

Based on protein’s KEGG pathway annotations, proteins were classified into different KEGG categories. A two-tailed Fisher’s exact test was performed to detect the significance of enrichment. If the *p*-value < 0.05, the KEGG category was considered to be significantly enriched.

Based on protein domain annotation, proteins were classified into different domain categories. Fisher’s exact test was performed to detect the significance of domain enrichment. If the *p*-value < 0.05, the protein domain was considered to be significantly enriched.

To further hierarchical clustering analysis, we first filtered the *p*-value to preserve the entries in any comparison group with a *p*-value less than 0.05. Based on the function x = −log10 (*p*-value), we transformed the filtered *p*-value matrix to an x matrix, and then a z-transform was performed on the x values of each entry. Finally, the z scores were clustered using the one-way hierarchical clustering method. 

The R package heatmap from gplots [41] was used to visualize the cluster membership by heat map.

### 2.9. Protein–Protein Interaction Network

The protein–protein interaction analysis of differentially expressed proteins was performed using STRING database (https://string-db.org, accessed on 25 November 2022). If confidence score ≥ 0.7, the protein–protein interaction was considered to be of high confidence. The piece of software Cytoscape (v3.9.1) [42] was used to visualize the protein–protein interaction network.

### 2.10. Transcriptome Analysis

The transcriptome data came from our previously published paper [35]. Transcriptome differential expression analysis was performed using the R package DEseq2. The heat map was drawn using the R package heatmap from gplots [41].

## 3. Results

### 3.1. Stage Definition of Seed Germination and Seedling Growth

According to the definition of Bewley, seed germination is divided into three stages: rapid water imbibition, limited water absorption, and the embryonic axis elongation mark the accomplishment of an increase in water uptake, followed by seedling growth [1]. Previously, we labelled the dry seed as stage 1, the rapid increase in water uptake as stage 2, the slow increase in water uptake as stage 3, the hypocotyl extension as stage 4, the cotyledon unfolding as stage 5, and the four-true-leaf unfolding as stage 6 [35]. Among the six stages, stages 1 to 4 represent the seed germination process, and stages 4 to 6 represent the seedling growth process. 

In previous studies, we found that there were the most differentially expression genes of the process from stage 3 to stage 5 in *T. hispida* [35], accompanied by dramatic morphological changes (Appendix A). Therefore, stage 3, stage 4, and stage 5 were selected for label-free quantitative proteome analysis to probe the changes in protein profiles during seed germination and seedling growth.

### 3.2. Profile of the Proteome in Seed Germination and Seedling Growth

In total, 373,687 MS spectrums, 57,379 matched spectrums, 22,828 peptides, 22,030 unique peptides, and 6880 protein groups were identified from the germinated seeds and seedling growth at different stages (stage 3, stage 4, and stage 5) in *T. hispida* (Figure 1a). A RRM domain-containing protein (Fragment) with a molecular mass of 9.4 kDa and auxin transport protein BIG with a molecular mass of 567.3 kDa were defined as the extreme results of the molecular mass. The molecular masses of all identified proteins mainly fell in the range of 10–60 kDa (5881 proteins, 85.48%): 10–20 kDa (1986 proteins, 28.87%), 20–30 kDa (1600 proteins, 23.25%), 30–40 kDa (1083 proteins, 15.74%), 40–50 kDa (730 proteins, 10.61%), and 50–60 kDa (482 proteins, 7.01%) (Figure 1b, Appendix A). The distribution of peptide quantity per protein showed that the protein number reduced as the matching peptides increased in number (Figure 1c).

To obtain insights into the functions of the 6880 proteins, the COG/KOG, Gene Ontology (GO), Kyoto Encyclopedia of Genes and Genomes (KEGG), and Pfam annotations were performed. The 4966 identified proteins (72.18%) were subjected to COG/KOG annotation. Then, 2860 proteins (41.57%) were subjected to CO annotation, 2747 proteins (39.93%) were subjected to KEGG annotation, and 3080 proteins (44.77%) were subjected to Pfam annotation (Figure 1d, Appendix A).

### 3.3. Comparative Analysis of DAPs and DEGs in Seed Germination and Seedling Growth

After protein quantitative analysis, pairwise differentially abundant proteins (DAP) analysis was performed with the threshold of absolute value fold change ≥ 2.0 and coefficient of variation (CV) < 0.1 to investigate the protein bases of seed germination and seedling growth. Among these 6880 proteins, 2911 were defined as DAPs, including 87 for stage 3 vs. stage 4 (53 upregulated, 34 downregulated), 2280 for stage 4 vs. stage 5 (1154 upregulated, 940 downregulated), and 2094 for stage 3 vs. stage 5 (1205 upregulated, 1075 downregulated) (Figure 2a). Among stages 3, stage 4, and stage 5, differential expression analysis of pairwise stage was performed. A total of 30,874 differently expressed unigenes (DEGs) were identified. Further comparison of the DEGs and proteins revealed that 3126 DEGs produce proteins (Figure 2b), including 2015 DEGs of stage 4 vs. stage 3 (1417 upregulated, 598 downregulated), 1,589 DEGs of stage 5 vs. stage 4 (702 upregulated, 887 downregulated), and 2044 DEGs of stage 5 vs. stage 3 (1309 upregulated, 735 downregulated) (Figure 2c).

Seed germination is a short-term process, taking only 19 h from stage 3 to stage 4 in *T. hispida*. Combined transcriptomic and proteome analysis was performed. There were 2015 DEGs but only 89 DAPs between stage 4 and stage 3, indicating that regulation at the transcriptome level has been initiated during the process of stage 3 and stage 4. Further comparison of the 1417 upregulated DEGs in stage 4 vs. stage 3 and the 1154 upregulated DAPs in stage 5 vs. stage 4 revealed 523 common unigenes. Meanwhile, a comparison of the 598 downregulated DEGs in stage 4 vs. stage 3 and 940 downregulated DAPs in stage 5 vs. stage 4 revealed 208 common unigenes (Figure 2d). The hypergeometric test was used to perform the significance testing, and the *p*-values of the number of common unigenes were 5.88 × 10^−21^ and 1.17 × 10^−13^ in the upregulated and downregulated comparison groups, respectively. The maximal protein abundances generally lagged behind those of the transcript. The above results suggest that the regulation of transcriptional levels during seedling growth (stage 4 to stage 5) is already initiated during the seed germination (stage 3 to stage 4), resulting in changes in protein levels during seedling growth.

### 3.4. Function Comparative Analysis of DAPs and DEGs in Seed Germination and Seedling Growth

GO analysis was conducted to classify the DAPs’ functions as biological processes (BPs), molecular functions (MFs), or cellular components (CCs). Among the 87 DAPs in the seed germination process (stage 4 vs. stage 3), 26 played a role in 11 different BP categories, 1 was classified into the MF category, and 4 were related to CC and classified into 3 categories (Figure 3a). Meanwhile, the GO classification of 2280 DAPs in the seedling growth process (stage 5 vs. stage 4) showed that 139 played a role in 17 different BP category, 17 were classified to 12 different MF categories, and 6 were related to CC classified into 2 categories (Figure 3b). Most DAPs were involved in BP category in both the seed germination process and the seedling growth process. Biological process analysis revealed that the DAPs related to “metabolic process” (55.2% and 64.1%), “cellular process” (44.8% and 64.1%), and “response to stimulus” (27.6% and 23.7%) accounted for the high percentages of DAPs in both the seed germination process and the seedling growth process. Moreover, the “biological regulation” (22.4%) and “regulation of biological process” (19.2%) categories applied to many of the DAPs in the BP category in the seedling growth process.

To investigate the main synthetic and metabolic pathways in seed germination and seedling growth, the DAPs and DEGs of each compared group were subjected to KEGG enrichment analysis. The results are shown in Appendix A. In the seed germination process (stage 4 vs. stage 3), the DAPs were significantly enriched in two pathways: “Porphyrin and chlorophyll metabolism” and “Fatty acid elongation”, which is consistent with the morphologic change from stage 3 to stage 4 (Figure 3c). Among 20 significantly enriched pathways of DAPs in the seedling growth process (stage 5 vs. stage 4), “Carbon metabolism” was the most significantly enriched pathway, consisting of 125 DAPs; and “Flavonoid biosynthesis” and “Phenylpropanoid biosynthesis”, which are important for seed germination and seedling growth in *T. hispida* [35], were significantly enriched in the seedling growth process (Figure 3d). Nevertheless, through KEGG enrichment analysis of DEGs, we found that “Flavonoid biosynthesis” and “Phenylpropanoid biosynthesis” were significantly enriched for both the seed germination process and the seedling growth process (Appendix A), and 815 (29.23%) of the proteins from DEGs between stage 4 and stage 3 and between stage 5 and stage 4 were in common (Figure 3e). As the main executor of cell function, proteins are the main effector molecules that finally participate in cellular biological processes. The results show that flavonoid biosynthesis is regulated at the transcriptional level in both the seed germination process and the seedling growth process, but at the protein level, it initiates regulation from the seedling growth process, indicating that there is a difference in the type of flavonoids synthesized in seed germination and seedling growth.

### 3.5. Dynamic Regulation of Key Proteins in Flavonoid Biosynthesis and Plant-Hormone Signal Transduction during Seed Germination and Seedling Growth

The phenylpropanoid biosynthesis pathway is an upstream branch of flavonoid biosynthesis. A total of 49 DAPs from the phenylpropanoid biosynthesis (ko00940) pathway during seedling growth (stage 5 vs. stage 4), such as phenylalaninammo-nialyase (PAL), 4-coumarate-CoA ligase (4CL), and peroxidase (E1.11.1.7), were overall upregulated at stage 5 (Appendix A). Flavonoid synthesis enzymes can be divided into two types, namely, early biosynthesis proteins (EBPs), which are responsible for the production of common precursors, and late biosynthesis proteins (LBPs), which are responsible for the eventual product [43]. A total of 24 DAPs were identified in the flavonoid biosynthesis (ko00941) pathway, including 18 EBPs and six LBPs (Figure 4a, Appendix A). Among the 18 EBPs, there were eight synthase families, namely, trans-cinnamate 4-monooxygenase (three proteins; C4H), shikimate *O*-hydroxycinnamoyltransferase (three proteins; HCT), 5-*O*-(4-coumaroyl)-d-quinate 3’-monooxygenase (one protein; C3′H), caffeoyl-CoA O-methyltransferase (three proteins; OMT), chalcone synthase (three proteins; CHS), chalcone isomerase (three proteins; CHI), naringenin 3-dioxygenase (one protein; F3H), and flavonoid 3’-monooxygenase (one protein; F3′H). Among the six EBPs, there were eight synthase families, including flavonol synthase (one protein; FLS), bifunctional dihydroflavonol 4-reductase (two proteins; DFR), leucoanthocyanidin reductase (one protein; LAR), and phlorizin synthase (two proteins; PGT1).

To investigate the expression pattern of the 24 DAPs involved in flavonoid biosynthesis, a clustering analysis of protein abundance divided these DAPs into three clusters (Figure 4b). Flavonoid biosynthesis branches out from the phenylpropanoid biosynthesis pathway via the rate-limiting enzyme CHS [47]. ThCHS-P-1 and ThCHS-P-3 of cluster3 were upregulated in stage 3 and stage 4, and ThCHS-P-2 of cluster 1 was upregulated in stage 5, indicating that the different ThCHS proteins are responsible for different stages of flavonoid biosynthesis. DFR, as an EBP, provides one entryway to anthocyanin biosynthesis [48], and all ThDFR (ThDFR-P-1 and ThDFR-P-2) proteins were upregulated in stage 5, suggesting that anthocyanin biosynthesis was initiated in the seedling growth process. Furthermore, a total of 14 DAPs involved in the plant-hormone signal transduction (ko04075) pathway in *T. hispida*, including two GH3 proteins, were involved in the auxin pathway—one ARR-B in the cytokinine pathway, three PYL proteins and one SNRK2 protein in the abscisic acid pathway, one EIN2 protein in ethylene pathway, and four BSK proteins and one TCH4 proteins in the brassinosteroid pathway (Appendix A). The proteins abundance heatmap shows that most of the 14 DAPs were upregulated in the cotyledon-unfolding period (stage 5) (Appendix A), indicating that the five hormones are important for the transition from seed germination to seedling growth in *T. hispida*.

### 3.6. The Functional Differences of DAPs in Seed Germination and Seedling Growth

To investigate differences in the functions of DAPs between seed germination and seedling growth, the clustering analyses of KEGG and GO enrichment results were performed. The clustering analysis of KEGG enrichment results indicated that most of the biosynthesis and metabolism pathways were significantly associated with the seedling growth process (stage 5 vs. stage 4), such as “Flavonoid biosynthesis”, “Carotenoid biosynthesis”, “Stilbenoid, diarylheptanoid and gingerol biosynthesis”, “Isoquinoline alkaloid biosynthesis”, “Glucosinolate biosynthesis”, “Phenylpropanoid biosynthesis”, “Tropane, piperidine and pyridine alkaloid biosynthesis”, “Starch and sucrose metabolism”, and “Glycerolipid metabolism” (Figure 5a). Meanwhile, these biosynthesis and metabolism proteins of secondary products were upregulated from stage 4 to stage 5 (Figure 5b). The results suggests that the biosynthesis and metabolism of a large number of secondary products may be of great significance for the seedling growth of *T. hispida*.

The clustering analysis of GO enrichment results during seed germination and seedling growth showed that eight GO terms of the biological process were significantly associated with the seed germination process (stage 4 vs. stage 3) (Figure 5c, Appendix A), among which, the “hyperosmotic salinity response” (GO:0042538) term is any process that results in a change in state or activity of a cell or an organism as a result of detection of an increase in the concentration of salt in the environment, and which is upregulated in seed germination process (Figure 5d, Appendix A), indicating that the seed germination process plays an important role in the formation of salt and alkali resistance in *T. hispida*. Except for three GO terms (“epithelial cell differentiation”, “ribonucleoprotein complex subunit organization”, and “shoot system development”), all of the GO terms of the biological process were upregulated in the seedling growth process (stage 5 vs. stage 4) (Figure 5d, Appendix A). Among these upregulated GO terms of biological process, the “cellular response to water stimulus”, “response to water deprivation”, and “cellular response to water deprivation” terms are the processes that resulted in changes the in state or activity of a cell as a result of a water deprivation stimulus—prolonged deprivation of water (Appendix A)—indicating that the seedling growth process (from stage 4 to stage 5) is important for the tolerance to drought stress in *T. hispida*. Furthermore, the clustering analysis results of the protein domains, GO terms for molecular function, and GO terms for the cellular components are shown in Appendix A, respectively.

### 3.7. The Abundance-Pattern Analysis of DAPs in Seed Germination and Seedling Growth

A total of 79 DAPs were involved in the “response to water deprivation” in *T. hispida*, and the protein abundances of the 79 DAPs are displayed in a cluster heatmap (Figure 6a, Appendix A). In Figure 6a, the relatively high and low protein abundances are represented by red and blue segments, respectively. The clustering analysis of the 79 DAPs between the three samples show a clear clustering pattern. Cluster 1 contains 28 DAPs that were upregulated in stage 3 and stage 4, suggesting that these proteins are involved in the water-deprivation response in the seed germination process. Cluster 2 contains 2 DAPs that were upregulated in stage 4 and stage 5. Cluster 3 contains the largest number of DAPs (49) that were uniquely upregulated in stage 5, indicating that these proteins are involved in the water-deprivation response at seedling growth. Meanwhile, 15 DAPs were identified in the “hyperosmotic salinity response” term and were further divided into three clusters, among which cluster 1, cluster 2, and cluster 3 were found to be upregulated in stages 3 and 4, stages 4 and 5, and stage 5, respectively (Figure 6b, Appendix A). The results showed the DAPs that respond to water deprivation and respond to hyperosmotic salinity have three different abundance patterns, suggesting that there were different proteins in the seed germination process and seedling growth process to play the function of resisting hyperosmotic salinity and water deprivation stress.

Comparative analysis showed that eight DAPs responded to both water deprivation stress and hyperosmotic salinity stress (Figure 6c), including two P5CSs (delta-1-pyrroline-5-carboxylate synthetase), one Y3IP (Ycf3-interacting protein), one low-temperature-induced 65 kDa protein-like molecule, and four PRXs (Peroxidase). The proline acts as a compatible osmolyte to counteract salinity and drought [49]. Previous studies have reported that proline biosynthesis is a two-step process. The first-step process is catalyzed by the rate-limiting enzyme of P5CS in most plant species, the ThP5CS-P-1 (TRINITY_DN789_c0_g2_i19.p1) is upregulated in stage 5, and the ThP5CS-P-2 (TRINITY_DN12736_c1_g1_i3.p1) is upregulated in stage 4 and stage 5 (Figure 6a,b, Appendix A). Peroxidases play a variety of biological roles, including plant defense responses to salt and drought stresses [50]. In the *T. hispida*, four PRXs that responded to both salt and drought stresses were all upregulated in stage 5 (Figure 6a,b, Appendix A). Meanwhile, Y3IP enhanced the plant’s tolerance to salt stress by reducing ROS levels and hastening lateral root growth through auxin biosynthesis and transport pathways [51]. ThY3IP-P was upregulated in stage 5 (Figure 6a,b, Appendix A). The results indicate that several mechanisms exist to enhance the tolerance to salinity and drought during the seedling growth process in *T. hispida*.

A protein–protein interaction network (PPI) analysis was performed to investigate the interaction relationship between the eight DAPs that responded to both salinity and drought and other DAPs. A total of 38 DAPs were involved in the interaction network, and clustering analysis of the 38 DAPs showed two clear clustering patterns (Appendix A). Custer 1 and cluster 2 were regulated in stages 3, 4, and 5, respectively. The interaction network is shown in Figure 6d. A lot of interacting DAPs fit the abundance pattern of cluster2.

## 4. Discussion

To the best of our knowledge, this study was the first to investigate the protein-abundance profile of *T. hispida* during the transition process from seed germination to seedling growth using label-free proteomics approaches. This study also revealed the advantages of proteomics in the study of short-term development processes, such as seed germination, by comparing proteomics with transcriptomics. The proteins are the ultimate biomolecules that complement the structures and functions of living systems [25]. Previously, we found that there were significant differences in transcriptional expression levels of the hypocotyl-extension period (stage 3 vs. stage 4) and cotyledon-unfolding period (stage 4 vs. stage 5) via comparative transcriptomic analysis, among which, the hypocotyl-extension period had the most DEGs [35]. However, the expression status of genes obtained by transcriptomics analysis reflected neither the correct number of proteins nor their regulatory status [52]. Label-free quantitation proteomics allows us to attain complete knowledge of ongoing cellular processes [53]. In the present study, a total of 6880 protein groups were identified from the various stages of seed germination to seedling growth in *T. hispida* (Figure 1a). The molecular masses of all identified proteins mainly fall within the range of 10–60 kDa (5881 proteins, 85.48%) (Figure 1b). The quantitative analysis showed that quantitative proteins constituted approximately 88.5% of the identified proteins (Appendix A). Comparative analysis showed that 87 (53 upregulated, 34 downregulated) and 2280 (1154 upregulated, 940 downregulated) DAPs were identified in stage 3 vs. stage 4 and stage 4 vs. stage 5, respectively (Figure 2c). In contrast to the transcriptomic results, differential protein analysis revealed that there were more differential proteins in stage 4 vs. stage 5 than in stage 3 vs. stage 4 (Figure 2a,c), which is more consistent with morphological changes in *T. hispida* (Appendix A). The maximal protein abundances generally lag those of transcripts. Thus, the results of this differential proteomics analysis imply that proteomics may be better for studying short-term developmental processes, such as seed germination.

The results of GO classification and KEGG enrichment analysis of the DAPs in seed germination (stage 4 vs. stage 3) and seedling growth (stage 5 vs. stage 4) suggest many changes in secondary metabolites, such as flavonoid and phenylpropanoid, as previously published for *T. hispida* (Figure 3) [35]. Flavonoids, which are secondary metabolites that exist widely in plants, are involved in plant responses to abiotic and biotic stresses [54,55]. Studies have shown that flavonoids regulate drought and salt tolerance in seed germination and seedling growth of plants by removing reactive oxygen species (ROS) [55,56]. In the present study, 18 EBPs and 6 LBPs involved in the flavonoid biosynthesis pathway were identified (Figure 4, Appendix A). The clustering analysis of protein abundance divided the 24 DAPs into three clusters (Figure 4), and multiple abundance patterns of ThCHS protein indicate that different ThCHS proteins are responsible for different stages of flavonoid biosynthesis in *T. hispida*. Meanwhile, the upregulation of all ThDFR proteins in stage 5 suggests that anthocyanin biosynthesis is initiated at the time of seedling growth and may directly affect the color change during the four-true leaf unfolding phase [35].

The *T. hispida* is a woody halophyte that is extremely resistant to drought and soil salinity. The research on its drought and salt resistance mechanisms is beneficial to promote the development of high resistance crop breeding [57,58,59]. Meanwhile, the seed germination process and seedling growth process are highly related to subsequent vegetative growth, and therefore directly affect crop yield and quality. Therefore, in this study, we used *T. tamarix* as study material to reveal the molecular mechanisms of drought and salt resistance during seed germination and seedling growth through dynamic protein profiling. Clustering analyses of the GO enrichment of DAPs during seed germination and seedling growth showed that the “hyperosmotic salinity response” term is significantly associated with the seed germination process, indicating that the seed germination process plays an important role in the formation of salinity resistance in *T. hispida*. Meanwhile, the “cellular response to water stimulus”, “response to water deprivation”, and “cellular response to water deprivation” terms were significantly associated with the seedling growth process, indicating that the seedling growth process is important for the tolerance to drought stress in *T. hispida* (Figure 5c,d).

In *T. hispida*, 79 and 15 DAPs were involved in the “response to water deprivation” and “hyperosmotic salinity response”, respectively. Furthermore, two P5CSs, one Y3IP, one low-temperature-induced 65 kDa protein-like molecule, and four PRXs responded to both water deprivation stress and hyperosmotic salinity stress. As a common osmolyte, proline has many effects on plants under drought stress, such as sunflower seeds accumulating more proline in response to drought or salt stress [60]. In plants, P5CS and Δ1-Pyrroline-5-carboxylate reductase catalyze the glutamic acid-to-proline reaction [61]. Under osmotic stress, the accumulation of proline in plants is regulated primarily by P5CS; for example, studies on rice and other plants have shown that the proline content is increased by the upregulation of P5CS expression and activity under drought and salt stress [44,45,46]. In the present study, we found that ThP5CS-P-1 is upregulated in stage 5 and ThP5CS-P-2 is upregulated in stages 4 and 5, suggesting that drought and salt resistance is enhanced by upregulation of P5CS to increase proline content during seed germination and seedling growth in *T. hispida* (Figure 6a,b, Appendix A). In addition, PRXs and Y3IP also enhance drought and salt resistance during *T. hispida* seed germination and seedling growth through other mechanisms (Figure 6a,b, Appendix A). These results support the existence of several mechanisms that enhance tolerance to salinity and drought stress during seedling growth in *T. hispida*.

## 5. Conclusions

The significance of our research is that it is the first to provide insight into the proteomic changes in *T. hispida* seed germination and seedling growth using label-free based proteomics and to reveal the potential mechanisms responsible for the formation of drought and salt resistance in *T. hispida*. Through a comparative analysis of transcriptomics and proteomics, we found that proteomics may be better for studying short-term developmental processes, such as seed germination and seedling growth. Clustering analyses of proteomic GO enrichment results in *T. hispida* revealed that 79 DAPs were enriched in response to water deprivation and were upregulated during seedling growth, and 15 DAPs were enriched in hyperosmotic salinity response and were upregulated during seed germination. Further analysis shows that two P5CSs, one Y3IP, one low-temperature-induced 65 kDa protein-like molecule, and four PRXs were involved in both water deprivation and hyperosmotic salinity responses during *T. hispida* seed germination and seedling growth. This study has provided important insights into the dynamic proteome changes and drought and salt resistance formation underlying *T. hispida* seed germination and seedling growth.

## Figures and Tables

**Figure 1 genes-14-00656-f001:**
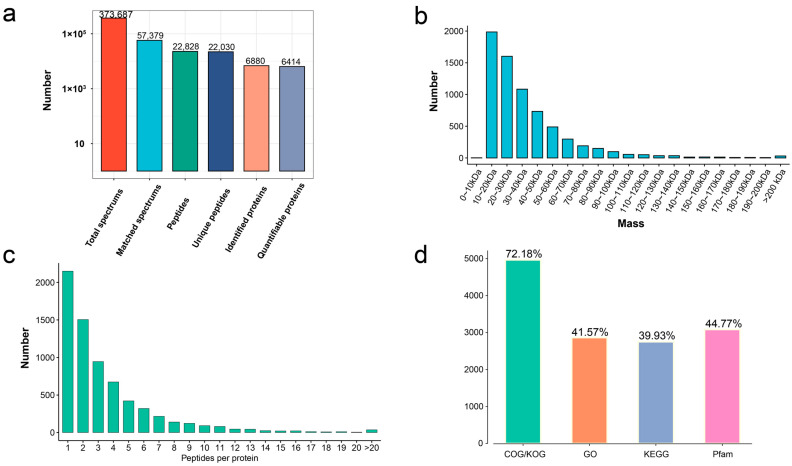
The distribution of proteome data and annotations. (**a**) Overview of protein identification. (**b**) Length distribution of identified peptides. (**c**) Peptide count distribution of identified proteins. (**d**) Protein functional annotation overview of COG/KOG, Gene Ontology, KEGG pathways, and protein domains.

**Figure 2 genes-14-00656-f002:**
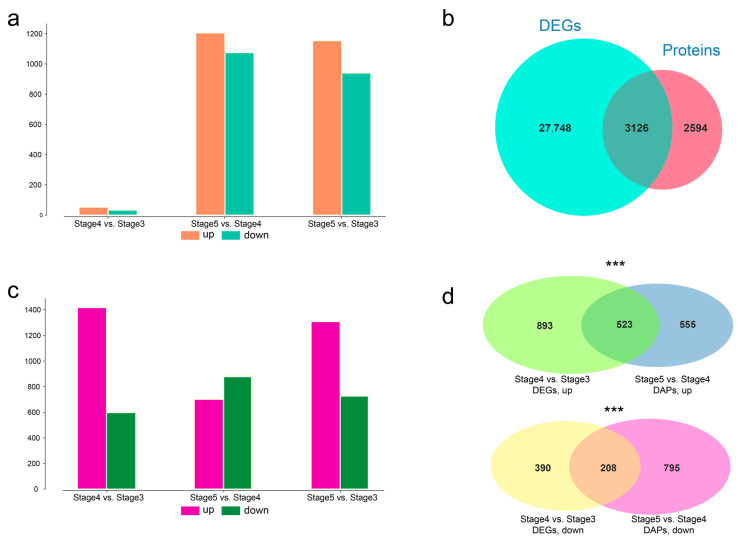
Comparative analysis between differentially abundant proteins (DAP) and differently expressed unigenes (DEGs). (**a**) Numbers of upregulated and downregulated DAPs at pairwise stages. (**b**) Venn plot between DEGs and identified proteins. (**c**) Numbers of upregulated and downregulated DEGs at pairwise stages. (**d**) Venn plot between DEGs of stage 4 and stage 3 and DAPs of stage 5 vs. stage 4. Upregulated DAPs and DEGs are shown at the top, and downregulated DAPs and DEGs are shown at the bottom. Asterisks indicate degree of significance (*** *p* < 0.001).

**Figure 3 genes-14-00656-f003:**
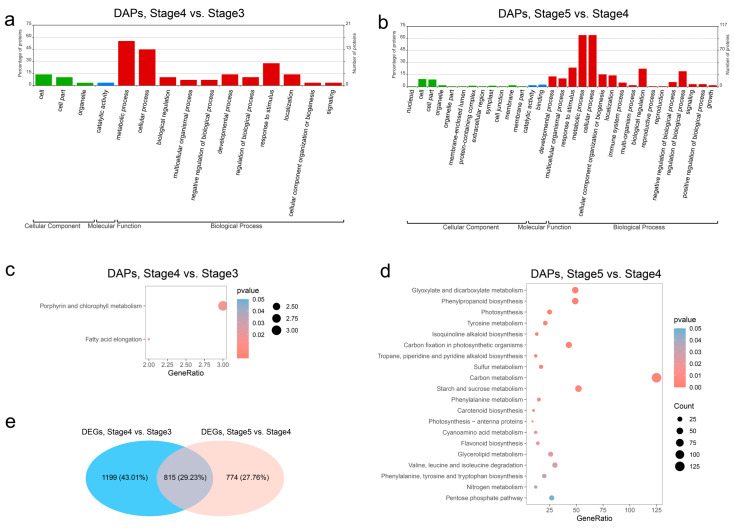
Functional analysis of differentially abundant proteins (DAP). (**a**) GO classifications of DAPs between stage 4 and stage 3. (**b**) GO classifications of DAPs between stage 5 and stage 4. (**c**) KEGG pathways enrichment of DAPs between stage 4 and stage 3. (**d**) KEGG pathways enrichment of DAPs between stage 5 and stage 4. (**e**) Venn plot between DEGs of stage 4 vs. stage 3 and DEGs of stage 5 vs. stage 4.

**Figure 4 genes-14-00656-f004:**
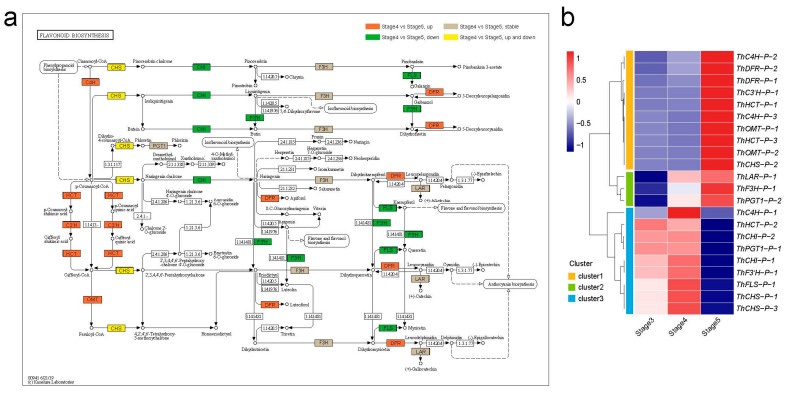
Analysis of DAPs in the flavonoid biosynthesis pathway. (**a**) Mapping of enriched DAPs in the flavonoid biosynthesis pathway (ko00941) [44,45,46]. The orange marks represent DAP upregulated in stage 5 vs. stage 4; the green marks represent DAPs downregulated in stage 5 vs. stage 4; the yellow marks represent the upregulated DAPs and downregulated DAPs in stage 5 vs. stage 4. The grey marks represent proteins stable in stage 5 and stage 4. (**b**) Abundance pattern of DAPs involved in flavonoid biosynthesis pathway at different stages.

**Figure 5 genes-14-00656-f005:**
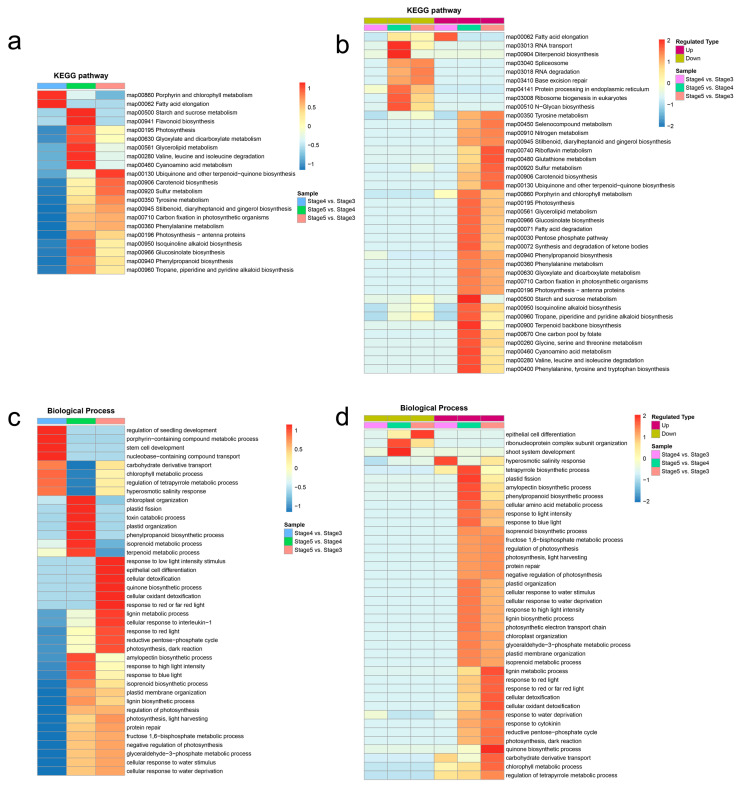
Clustering analyses of KEGG and GO enrichment results. (**a**) Clustering analysis of KEGG enrichment categories of DAPs. (**b**) Clustering analysis of KEGG enrichment categories of upregulated DAPs and downregulated DAPs. (**c**) Clustering analysis of Gene Ontology (GO) enrichment entries of DAPs. (**d**) Clustering analysis of GO enrichment entries of upregulated DAPs and downregulated DAPs. The horizontal side of the heatmap is different comparison groups, and the vertical side is the description of the correlation functions enriched by differentially abundant proteins in the different comparison groups. The color block corresponds to the functional descriptions of enrichment differentially abundant proteins, showing the degree of enrichment. Red means strong enrichment, blue means weak enrichment.

**Figure 6 genes-14-00656-f006:**
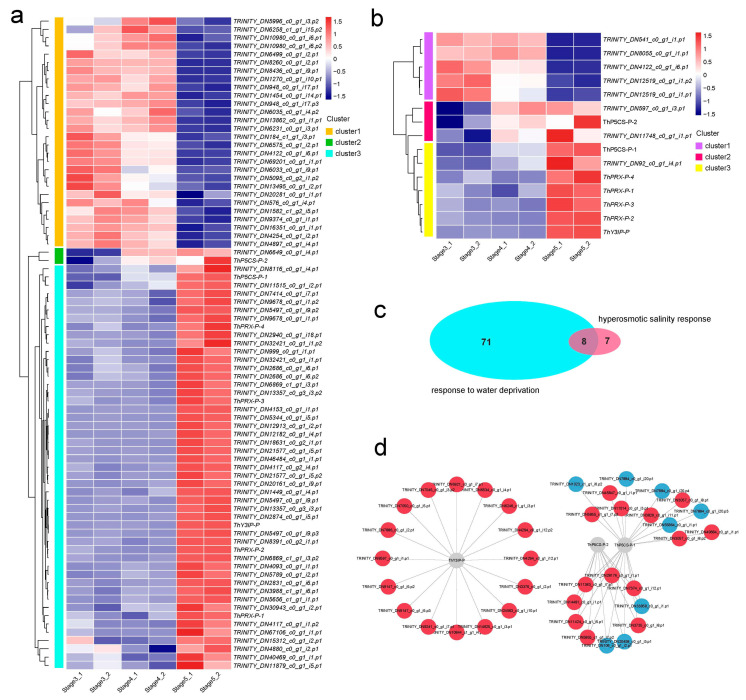
Abundance analysis of DAPs based on proteomics. (**a**) Heat map of water-deprivation-response-associated DAPs. (**b**) Heat map of hyperosmotic-salinity-response-associated DAPs. (**c**) Venn plot between response to water-deprivation-associated DAPs and hyperosmotic-salinity-response-associated DAPs. (**d**) Analysis of the functional network by STRING 11.0 of DAPs tied to responses to both water deprivation and hyperosmotic salinity. Red indicates significantly upregulated DAPs, and blue indicates significantly downregulated DAPs.

## Data Availability

Not applicable.

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
