# Peer review of "Proteome Dynamics Analysis Reveals the Potential Mechanisms of Salinity and Drought Response during Seed Germination and Seedling Growth in Tamarix hispida"

_genes, 2023, doi:10.3390/genes14030656_

Round 1
Reviewer 1 Report
This is a comprehensive study on the proteome dynamics of seed germination and seeding growth in Tamarix hispida plants. The authors collected samples from 3 stages presenting germination and seedling developments. The MS is well written but below issues should be clarify.
1. Germination conditions are not clear. Please provide more information about germination conditions such as temprature or presence of salt. Did you treat seeds with salt or not?
2. SDS-PAGE should be provided at least as a supplementary files.
3. Transcriptome data mentioned in the abstract and discussion sections should be explained more. Did you perform any transcriptome analysis?
Reviewer 2 Report
1. Title of the manuscript is not representing the whole content of the research, please revise. Because you relate your results with the drought and salinity stress.
2. Line 2: “seeding” should be “seedling”
3. For all the Figures in the manuscript, please increase the Font in the Graphics to become readable for the readers.
4. Additional comments:
a. What is the main question addressed by the research? Yes, the main question is addressed by the research.
b. Is it relevant and interesting? Yes, it is relevant and interesting but need to improve the information in the introduction section.
c. How original is the topic? The topic is original.
d. What does it add to the subject area compared with other published material? The subject is advance compared with other published material.
e. Is the paper well written? Yes, need improvement how they display the results.
f. Is the text clear and easy to read? Yes.
g. Are the conclusions consistent with the evidence and arguments presented? Can be improved.
h. Do they address the main question posed? Yes, but need more focus to the main question.
Reviewer 3 Report
The following MINOR REVISIONS are required before acceptance of the manuscript:
(1) Line No. 15: No clarity on 'perennial shrubs and trees'
(2) Line No. 34: Revise the word 'of' with 'like'
(3) Line No. 50: Rewrite as 'is highly tolerant'
(4) Line No. 88-89: Rewrite the statement for more clarity.
(5) Present standard error plot in the graphs
